# SEMANTIC-AWARE DIFFUSION MODEL FOR SEQUENTIAL RECOMMENDATION

## ABSTRACT

Sequential recommendation aims to predict the next click for a particular user based on their historical interacted item sequences. Recently, diffusion-based methods have achieved the state-of-the-art performance in sequential recommendation. However, they fail to effectively utilize the rich semantic information embedded in items during the diffusion process to accurately guide the generation, leading to sub-optimal results. To address this limitation, we designed SDREC, a **S**emantic-aware **D**iffusion model for sequential **Rec**ommendation. Our model introduces a novel architecture, the `Semantic Fusion Layer`, which leverages the embedding table from the encoder to incorporate item semantics into the diffusion process through an attention mechanism. Together with the well-designed contrastive and generative losses, SDREC effectively utilizes the item semantics in diffusion model, unleashing the potential of sequential recommendation. Our experiments show that SDREC has over $10\%$ relative gain with superior efficiency compared with existing methods.

## 1 INTRODUCTION

Sequential recommendation aims to mine the user's behavior patterns from historical interaction sequences and predicts the next item that the user is most likely to click in the future. It has attracted widespread attention due to its high commercial value in many business scenarios, such as streaming media (Covington et al., 2016), e-commerce (Chen et al., 2019), and social networking (Zhou et al., 2018). Since sequential recommendation needs to identify the most suitable item from the existing item set based on a user's historical interactions, the effectiveness of the recommendations hinges on the deep understanding of the semantics of the items (e.g., the categories that a movie belongs to) and the modeling of user interests (e.g., what kinds of movies does the user like).

In order to efficiently capture the item semantics and user preferences, various types of methods have been proposed for sequential recommendation (Hidasi & Karatzoglou, 2018; Yuan et al., 2019; Sun et al., 2019; Xie et al., 2021; Ren et al., 2020). Recently, diffusion model (Ho et al., 2020) has shown remarkable results in generation tasks from Computer Vision (CV) (Dhariwal & Nichol, 2021; Rombach et al., 2022) and Natural Language Processing (NLP) (Li et al., 2022; Gong et al., 2022). It defines a sequence of Gaussian distributions (i.e., Markov chain) instead of a single one in VAEs, granting it powerful fitting capabilities (Sohl-Dickstein et al., 2015; Vahdat & Kautz, 2020). Moreover, it addresses the training instability in adversarial learning in GANs (Salimans et al., 2016), making it easier to converge. Diffusion model corrupts inputs with random noises iteratively in the forward process. Under the guidance of some conditions, it can learn the distribution more deeply by removing noise and reconstructing the input. Thanks to its strong ability to fit complex distributions and to generate diverse outputs, it has been achieved the state-of-the-art performance in sequential recommendation (Wang et al., 2023; Li et al., 2023; Yang et al., 2024).

Despite their advancements, there remain some problems for existing diffusion recommenders. Diffusion model was initially designed for generative tasks, allowing the model to create content randomly to some degree as long as it satisfies the provided conditions. In contrast, recommendation tasks require precise retrieval of suitable items that a user is likely to click in the future (Lin et al., 2023), which requires a thorough understanding of the semantics of each item. This highlights the need to effectively integrate item semantics at each diffusion step to accurately guide the generation process. Unfortunately, existing diffusion recommenders rely solely on user preferences as conditions

(i.e., historical interaction sequences), resulting in a limited view of item semantics. What's worse, these methods introduce noise into user sequences and then pass them to Transformers or MLPs for denoising (Wang et al., 2023; Bénédict et al., 2023; Li et al., 2023), implicitly attempting to learn item semantics during the reverse diffusion process (see Figure 1(a)). The presence of noise in the input sequences hinders the model's ability to accurately capture the semantic relationships between items. Consequently, these methods struggle to capture the various semantics of items (e.g., a movie may belong to multiple categories). This limitation prevents them from effectively modeling users' diverse and dynamic interests, since user interests are indicated by user historical clicked items. As a result, they primarily identify simplistic features like historical click patterns as Figure 2(e) shows, leading to sub-optimal results. Therefore, how to efficiently leverage the item semantics in diffusion model becomes a vital problem for improved diffusion recommenders.

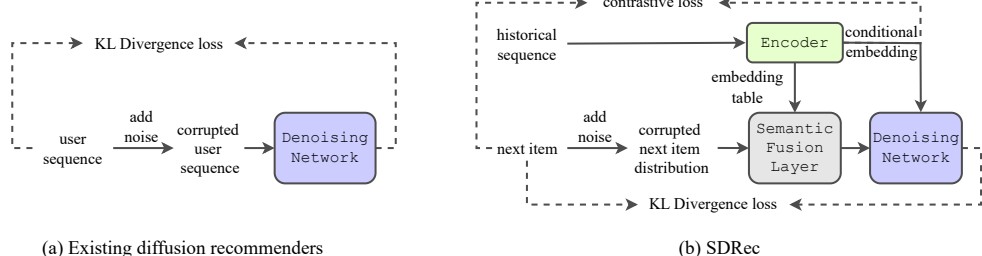

(a) Existing diffusion recommenders          (b) SDRec

Figure 1: **Comparison of diffusion recommenders.** (a) Existing methods, corrupt the user sequences by adding noise and then feed them to a denoising network, using KL Divergence loss to implicitly learn item semantics. (b) SDRec, explicitly learn item semantics via contrastive loss in the encoder and enhance the corrupted item distribution by leveraging the embedding table from the encoder, which injects semantic information through the `Semantic Fusion Layer`.

To address the problems mentioned above, we propose a novel **S**emantic-aware **D**iffusion model for sequential **Rec**ommendation named SDREC. The model is structured around an Encoder-Decoder architecture (refer to (b) in Figure 1). The encoder receives clean user sequences and explicitly capture the semantic relationships between items by contrastive learning. It also discerns user preferences from historical interactions, generating a conditional signal that effectively encodes user interests. Guided by this conditional signal, the decoder iteratively recovers the next item distribution from a noisy one. Before passing the noisy item distribution to the denosing network, we introduced a `Semantic Fusion Layer` that leverages the semantic embedding table from the encoder to transfer semantic information of items into the input distribution. Inspired by the attention mechanism, the embedding table is weighted by the noisy input distribution, enriching the semantic context while preserving the inherent randomness in diffusion model. Therefore, the reverse process of diffusion model can refer to the rich semantics embedded in item embeddings to accurately guide the generation. To sum up, the contributions of this work are as follows:

- We propose a novel diffusion recommender SDREC, which can effectively utilize the item semantics through the `Semantic Fusion Layer`, where the noisy input distribution is enriched by the semantic information from the item embedding table, improving the accuracy of the reverse process in the diffusion model.
- SDREC adopts an Encoder-Decoder architecture. Equipped with well-designed contrastive and generative loss, it can efficiently learn the item semantics and model user preferences simultaneously.
- We conduct extensive experiments to demonstrate impressive improvements over the baselines. Meanwhile, our model is more efficient than baselines, which is favorable for online serving.

## 2 BACKGROUND AND RELATED WORK

### 2.1 SEQUENTIAL RECOMMENDATION

Given a user's historical interacted item sequence arranged in chronological order $i_1, i_2, \cdots, i_m$, sequential recommendation aims to capture the user preferences from that and forecasts the next

item $i_\star$ that the user is likely to engage with in the future. The success of this task depends on a deep understanding of item semantics and accurate modeling of user interests. Due to the sequence format of the user's historical interactions and the discriminative nature of this task (i.e., distinguish between items that users are interested in or not), combing sequential models with contrastive learning to capture the semantics of items and mine user interests becomes a straightforward idea. Such methods include Convolutions Neural Networks (CNN) (Tang & Wang, 2018; Yuan et al., 2019) and Recurrent Neural Networks (RNN) (Hidasi & Karatzoglou, 2018; Hidasi et al., 2015). Recent advances with Transformer-based methods (Kang & McAuley, 2018; Sun et al., 2019) have pushed their performance even further.

However, in practical scenarios, users' interests are dynamic and evolving over time (Sachdeva et al., 2019; Li et al., 2023). To capture such diversity and uncertainty of user behaviors, generative models have been introduced for sequential recommendation, such as VAE-based (Sachdeva et al., 2019; Xie et al., 2021) and GAN-based (Bharadhwaj et al., 2018; Ren et al., 2020) methods. These kinds of approaches can also stimulate new interests for users and discover more business opportunities. However, these models suffer from intrinsic limitations such as the instability of GANs (Salimans et al., 2016) and the limited representation capacity of VAEs (Vahdat & Kautz, 2020). Such deficiencies hinder the deep modeling of complex user behaviors and item semantics.

## 2.2 DIFFUSION MODEL

Recent proposed diffusion model (Ho et al., 2020) mitigates the weaknesses of VAEs and GANs and push the state-of-the-art performance even further in generation tasks of both CV (Dhariwal & Nichol, 2021; Rombach et al., 2022) and NLP (Li et al., 2022; Gong et al., 2022). Inspired by non-equilibrium thermodynamics (Sohl-Dickstein et al., 2015), diffusion model defines a Markov chain consisting of $T$ forward diffusion steps, denoted as $x_{1:T}$, from an original distribution $x_0$. Specifically, in the forward process $q$ at step $t$, noise sampled from Gaussian distribution is added:

$$q(x_t|x_{t-1}) \sim \mathcal{N}(\sqrt{1-\beta_t}x_{t-1}, \beta_t \mathbf{I}), \tag{1}$$

where $\{\beta_t\}_{t=1}^{T}$ are a series of predefined parameters controlling the amount of noises added at each diffusion step. As $T \to \infty$, $x_T$ resembles an isotropic Gaussian distribution. Thanks to the Markov property, we can further calculate $x_t$ directly from $x_0$ with the following closed-form equation:

$$x_t = \sqrt{\bar{\alpha}_t}x_0 + \sqrt{1-\bar{\alpha}_t}\epsilon, \quad \epsilon \sim \mathcal{N}(\mathbf{0}, \mathbf{I}), \tag{2}$$

where $\alpha_t = 1 - \beta_t$, $\bar{\alpha}_t = \prod_{i=1}^{t} \alpha_i$. Using Bayes' theorem, the posterior distribution $q(x_{t-1}|x_t, x_0)$ can be derived from $\mathcal{N}(c_{1,t}x_0 + c_{2,t}x_t, \tilde{\beta}_t \mathbf{I})$, where $c_{1,t} = \frac{\sqrt{\bar{\alpha}_{t-1}}\beta_t}{1-\bar{\alpha}_t}$, $c_{2,t} = \frac{\sqrt{\alpha_t}(1-\bar{\alpha}_{t-1})}{1-\bar{\alpha}_t}$, $\tilde{\beta}_t = \frac{1-\bar{\alpha}_{t-1}}{1-\bar{\alpha}_t}\beta_t$. Thus, a neural network model can be used to fit $x_0$ and subsequently learn the reverse process $p_\theta$ for any step $t$:

$$p_\theta(x_{t-1}|x_t) \sim \mathcal{N}(c_{1,t}f_\theta(x_t, t, c) + c_{2,t}x_t, \tilde{\beta}_t \mathbf{I}), \tag{3}$$

where $f_\theta$ is the model with parameters $\theta$. Note that a classifier-free conditional diffusion model will further accept a conditional signal $c$ as input (Ho & Salimans, 2022). The model will be optimized by maximizing the variational lower bound of the log-likelihood of the input data $x_0$:

$$\mathcal{L} = \mathbb{E}_{x_0}\left[-\log p_\theta(x_0)\right] \leq \mathbb{E}_{x_0}\left[\sum_{t=1}^{T} \text{KL}\left(q(x_{t-1}|x_t, x_0)\|p_\theta(x_{t-1}|x_t)\right)\right] + C, \tag{4}$$

where $C$ is a constant independent of the model parameter $\theta$, KL is Kullback-Leibler Divergence.

## 2.3 DIFFUSION RECOMMENDERS

With the merits of diffusion model's tractability and strong representation capability (Sohl-Dickstein et al., 2015; Ho et al., 2020), recent studies have explored integrating diffusion models into sequential recommendation and achieved state-of-the-art performance (Wang et al., 2023; Li et al., 2023; Yang et al., 2024). For example, DiffRec (Wang et al., 2023) modifies the noise scale in diffusion model to ensure personalized recommendations, DiffuRec (Wang et al., 2023) injects uncertainty into item representations and reconstruct them by diffusion model in order to capture users' multi-level

interests, DCDR (Lin et al., 2023) proposes to use step-wise discrete operations to add noise during the diffusion process, RecFusion (Bénédict et al., 2023) adopts a binomial Markov diffusion process to fit the discrete recommendation datasets, DreamRec (Yang et al., 2024) proposes to generate the oracle item via diffusion model without any discriminative information.

Despite their success, these methods have not effectively leveraged item semantics during the diffusion process to generate high-quality recommendations. Originally, diffusion models were designed for generative tasks, allowing for some randomness in content creation as long as the provided conditions were met. However, recommendation tasks require the precise retrieval of items that are likely to engage the user (Lin et al., 2023), which necessitates a deep and comprehensive understanding of item semantics to guide generation accurately. Unfortunately, current diffusion recommenders rely solely on user historical sequences as conditions, lacking a global awareness of item semantics. Furthermore, the noise introduced to the user sequences impairs the model's ability to accurately capture the semantic relationships between items, further reduces the quality of recommendations.

To illustrate how item semantics affect the recommendation quality, we extract data from Movielens-1M (Harper & Konstan, 2015) as an example. Since category is an inherent attribute of each movie, we can use categories to express the semantics of movies. We count the number of each category that appears together with Drama movies (Category 6). As Figure 2(a) shows, Comedy, Romance, Drama, Action, and Thriller categories (Category 2, 5, 6, 7, and 9) often appear together, suggesting that Drama movie contains multiple semantics. Figure 2(b) shows the category counts of movies clicked by a user and (c) shows in a click timeline view. We can see that this user has a strong interest in Comedy, Drama, Action, and Thriller (Category 2, 6, 7, and 9), which aligns the semantic correlations confirmed from Figure 2(a). He also occasionally explores other categories like Adventure and Romance (Category 3 and 5), indicating his preferences are dynamic and diverse.

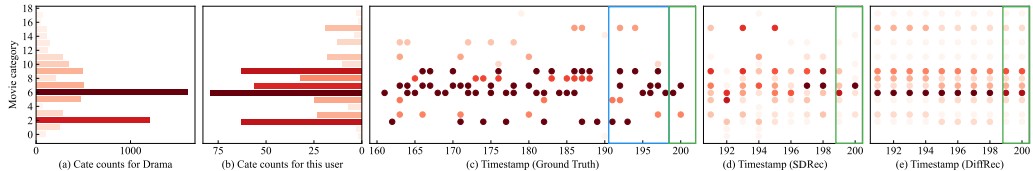

Figure 2: **Case study on how item semantics affects the recommendation results.** (a) The number of each category that appears together with Drama movie (Category 6). Darker red indicates stronger correlations with Drama. (b) The number of each category that the user clicks according to his historical sequence. Darker red categories denote user major interests. (c) Categories of the movie clicked by the user at each timestamp. (d) Predictions of SDREC for this user in category level. (e) Predictions of DiffRec (Wang et al., 2023) for this user in category level.

We applied a diffusion recommender DiffRec (Wang et al., 2023) to this case, shown as Figure 2(e). The corresponding ground truth is in the blue square (training data) and green square (validation and test data) in Figure 2(c). We collected top10 results predicted by this method at each click timestamp and accumulate them in the category level. The results indicate that the movie categories recommended by DiffRec are largely similar across timestamps without considering the diversity and dynamics of user interests.This is due to DiffRec's inability to fully learn and leverage item semantics (i.e., movie categories) during the diffusion process, resulting in an incomplete modeling of user interests, which are based on their interaction history. This also leads to inaccurate guidance during the generation process. As a result, it tends to recommend content mechanically based on past click patterns. However, user interests are diverse and dynamic, making this approach ineffective. For SDREC, it can efficiently learn and leverage item semantics through the `Semantic Fusion Layer` and produce high-quality recommendations. For example, after identifying Drama is the user's main interest, the model can recommend Adventure and Romance (Category 3 and 5) that are semantically related to Drama.

## 3 PROPOSED METHOD: SDREC

SDREC includes a semantic encoder and a denoising decoder, cooperating with the contrastive loss served for item semantic learning and the KL Divergence loss served for next item distribution

learning. It is designed to effectively integrate item semantics into the diffusion process for accurate generation.

## 3.1 MODEL DESIGN

Figure 3 shows the Encoder-Decoder architecture of SDREC. The semantic encoder serves to convert historical item IDs into embeddings and extract the semantic correlations of items, generating a $D$-dimensional conditional embedding $c$, which can also be considered as the representation of user interests. Based on the condition signal $c$ from the encoder, the decoder is utilized to reconstruct the distribution of the next item from a noisy input. Before the noisy input is passed to the denoising network, the embedding table from the encoder will be fed into the `Semantic Fusion Layer` to offer a global view of item semantics during the reverse diffusion process.

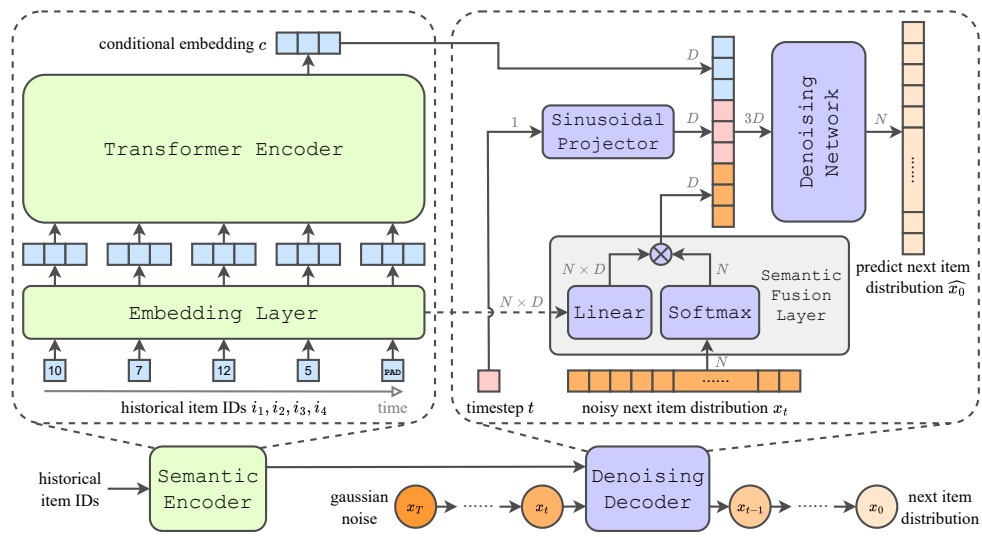

Figure 3: **Model architecture overview.** A semantic encoder is leveraged to encode historical sequences into a conditional embedding $c$. The decoder predicts the clean distribution of the next item $\hat{x}_0$ based on noisy distribution $x_t$ and conditional embedding $c$. Meanwhile, the `Semantic Fusion Layer` is designed to efficiently inject item semantics into the diffusion process.

**Semantic Encoder**   User's historical interacted items $i_1, i_2, \cdots, i_m$, which are sorted in chronological order, are firstly fed into an embedding layer (i.e., embedding lookup table), producing $m$ $D$-dimensional item embeddings $e_1, e_2, \cdots, e_m$. Because of the inconsistency of the length of users' historical sequences, we just consider the last $m$ items. Conversely, for sequences with less than $m$ items, padding tokens will be appended to reach the length of $m$. Then, a Transformer encoder (Vaswani, 2017) is applied to extract the semantic correlation of items based on their co-occurrence. Since no noise is injected into the inputs, the encoder can learn item semantics with greater accuracy. Finally, the last non-pad token embedding from the outputs of the last Transformer layer will be treated as the conditional embedding $c$. The above process can be described as below:

$$c = f_\varphi(i_1, i_2, \cdots, i_m). \tag{5}$$

Note that the conditional embedding $c$ can also be regarded as the tight representation of user interests.

**Denoising Decoder**   Following the classifier-free conditional diffusion model (Ho & Salimans, 2022), the decoder generates the distribution of the next item from a noisy one $x_t$, guided by the conditional signal $c$. Additionally, a `Semantic Fusion Layer` is designed to take advantage of accurate correlations and rich semantics of items from the encoder while preserving the inherent randomness in the diffusion model. Specifically, given a $N$-dimensional noisy distribution $x_t$, where $N$ is the total number of items in the dataset, we regard $x_t$ as the attention score in the traditional

attention mechanism (i.e., $QK^T$), and the value (i.e., $V$) is the detached embedding table $E$ from the embedding layer of the encoder, which is $N \times D$ dimension. Then, we can perform the attention mechanism by:

$$O_t = softmax(x_t) \times (W_v E), \tag{6}$$

where $W_v$ is the weight matrix in the linear layer. The output $O_t$ is a $D$-dimensional vector which can be considered as the weighted sum of all item embeddings according to the input noisy distribution.

By employing this layer, we can integrate global item semantics into the inputs of the denoising network, effectively addressing the issue of insufficient awareness of item correlations and semantics in the traditional diffusion recommenders. During the reverse diffusion process, the denoising network can refer to the rich semantics embedded in item embeddings to accurately guide the generation direction. Besides, the `Semantic Fusion Layer` compresses the distribution vector considering that typically $N \gg D$, thus diminishing the input size for the final denoising network and consequently reducing computational costs.

After that, we project the scalar diffusion timestep $t$ into a $D$-dimensional embedding by sinusoidal function (Vaswani, 2017), together with the conditional embedding $c$ and the attention output $O_t$, producing a vector that contains both item attributes and noise degree. Subsequently, this concatenated vector is passed through a denoising network (i.e., MLP) to derive a refined clean next item distribution, denoted as $\hat{x}_0$. The above process can be described as follows:

$$\hat{x}_0 = f_\theta(x_t, t, c, E). \tag{7}$$

With the comprehensive awareness of item semantics introduced by `Semantic Fusion Layer`, the decoder can reconstruct a more realistic and accurate distribution for the next item.

### 3.2 TRAINING PHASE

During training, we optimize the encoder and decoder simultaneously, with discriminative learning for the semantic encoder and generative learning for the denoising decoder.

**Discriminative Learning**   Compared to the generative methods, discriminate methods usually exhibit superior abilities to capture deterministic features (Bernardo et al., 2007). Thus, contrastive learning will be applied to the encoder output and the item embeddings, so that the embedding table $E$ will contain discriminant semantic information (e.g., semantic correlations). Specifically, we propose to align the conditional embedding $c$ with the next item embedding $e_\star$:

$$\mathcal{L}_D = \mathbb{E}_{(c,e_\star)} \left[ -\log \frac{\exp(c^T \cdot e_\star)}{\sum_{e \in E} \exp(c^T \cdot e)} \right]. \tag{8}$$

This loss minimizes the disparity between the output of the encoder and the embedding of the ground truth next item. Additionally, it brings similar items closer in the representation space, better reflecting their semantics and thus enabling the subsequent diffusion model to generate a more accurate distribution for the next item.

**Generative Learning**   During the generative learning, the decoder will recover the distribution of the next item based on the conditional signal $c$ and the item semantics injected by the `Semantic Fusion Layer`. In the forward process $q(x_t|x_{t-1})$, following the Eq.(2), Gaussian noise is added to the ground truth distribution $x_0$, which is the one-hot encoding of the ground truth next item $i_\star$. In the reverse process, instead of predicting the noise $\epsilon$, we predict the distribution itself (i.e., $\hat{x}_0$ in Eq.(7) ). Following Jin et al. (2023), we optimize the diffusion model by the KL Divergence loss:

$$\mathcal{L}_G = \mathbb{E}_{(x_0,c,t)} \left[ \text{KL} \left( x_0 \| f_\theta \left( x_t, t, c, E \right) \right) \right]. \tag{9}$$

This loss maximizes the probability of the ground truth by bringing $f_\theta(x_t, t, c, E)$ and $x_0$ closer. Since $\mathcal{L}_G$ focuses on optimizing the decoder, to stabilize the item representation during training, we use `detach` function to block the gradient propagation of the $\mathcal{L}_G$ to the embedding table $E$.

By integrating both the discrimination and generation training objectives, the comprehensive training loss of the entire model is the sum of $\mathcal{L}_D$ and $\mathcal{L}_G$. Besides, to alleviate overfitting, we randomly replace the conditional signal $c$ by a zero vector with probability $p_u$, which can be seen as unconditional training of the diffusion model. Algorithm 1 shows the details of the training phase.

---

**Algorithm 1** Training Phase

---

1: **repeat**
2:     Sample user historical sequence $i_1, \cdots, i_m, i_\star$ from training dataset $\mathcal{D}$.
3:     $c = f_\varphi(i_1, i_2, \cdots, i_m)$
4:     $e_\star = f_\varphi.\mathtt{Emb}(i_\star)$
5:     Compute $\mathcal{L}_D$ by Eq.(8).
6:     Sample $t \sim \mathrm{Uniform}(\{1, 2, \cdots, T\})$.
7:     $x_0 = \mathtt{OneHot\_Encode}(i_\star)$
8:     Compute $x_t$ by Eq.(2).
9:     $E = f_\varphi.\mathtt{Emb}.\mathtt{weight}.\mathtt{detach()}$
10:     With probability $p_u$: $c = \mathbf{0}$.
11:     Compute $\mathcal{L}_G$ by Eq.(9).
12:     Update $\varphi$ and $\theta$ via loss $\mathcal{L} = \mathcal{L}_D + \mathcal{L}_G$.
13: **until** converged

---

### 3.3 SAMPLING PHASE

In the sampling phase, the distribution of the next item will be recovered by the reverse denoising steps. Inspired by Yang et al. (2024), we first restrict the effect of the conditional signal $c$ at the beginning of denoising in order to provide more diverse results. We achieve this by designing a *reweight strategy* to modify the decoder outputs:

$$\tilde{f}_\theta(x_t, t, c, E) = \frac{1}{1+t} f_\theta(x_t, t, c, E) + \frac{t}{1+t} f_\theta(x_t, t, \mathbf{0}, E). \tag{10}$$

During the early denoising phase (i.e., $t = T$), higher $t$ limits the strength of the conditional signal $c$, avoiding undermining diffusion generalization. As the denoising step proceeds, gradually decreased $t$ will guide the model to generate outputs aligned with user interests effectively. Compared to Yang et al. (2024), our reweight strategy does not require the tuning of the hyper-parameter and thus has better adaptability.

Subsequently, following Eq.(3), the decoder will gradually recovers the distribution of the next item by the reverse process starting from a Gaussian noise $\tilde{x}_T \sim \mathcal{N}(\mathbf{0}, \mathbf{I})$, which can be reparameterized as follows:

$$\tilde{x}_{t-1} = \frac{\sqrt{\bar{\alpha}_{t-1}}\beta_t}{1 - \bar{\alpha}_t} \tilde{f}_\theta(\tilde{x}_t, t, c, E) + \frac{\sqrt{\alpha_t}(1 - \bar{\alpha}_{t-1})}{1 - \bar{\alpha}_t} \tilde{x}_t + \sqrt{\frac{1 - \bar{\alpha}_{t-1}}{1 - \bar{\alpha}_t}\beta_t}z, \quad z \sim \mathcal{N}(\mathbf{0}, \mathbf{I}), \tag{11}$$

where $t \in \{T, T-1, \cdots, 1\}$. Algorithm 2 shows the details of the sampling phase. Once the final predicted distribution $\tilde{x}_0$ is reconstructed, we firstly exclude items that have interacted within the user's historical sequence and then select the TopK items with the highest probabilities in $\tilde{x}_0$ to form the final recommendation list.

---

**Algorithm 2** Sampling Phase

---

1: Obtain user historical sequence $i_1, \cdots, i_m$ from test dataset $\mathcal{D}_t$.
2: $c = f_\varphi(i_1, i_2, \cdots, i_m)$
3: $E = f_\varphi.\mathtt{Emb}.\mathtt{weight}$
4: Sample $\tilde{x}_T \sim \mathcal{N}(\mathbf{0}, \mathbf{I})$.
5: **for** $t = T, T-1, \cdots, 1$ **do**
6:     Compute $\tilde{f}_\theta(\tilde{x}_t, t, c, E)$ by Eq.(10).
7:     Compute $\tilde{x}_{t-1}$ by Eq.(11).
8: **end for**
9: **return** $\tilde{x}_0$

---

## 4 EXPERIMENTS

In this section, we will demonstrate that SDREC exhibits superior recommendation capabilities compared to state-of-the-art baselines. Furthermore, we highlight the significance of our design

choices, including the `Semantic Fusion Layer` and contrastive learning, which play crucial roles in effectively integrating item semantics into diffusion model for enhanced recommendation.

## 4.1 EXPERIMENT SETUP

**Datasets** We use three real-world datasets to validate the performance of our model. 1) **Amazon Beauty** and 2) **Amazon Toys and Games** (He & McAuley, 2016; McAuley et al., 2015) are two categories of Amazon review datasets, which contain a collection of user-item interactions on Amazon. 3) **Movielens-1M** (Harper & Konstan, 2015) is a widely used benchmark dataset that includes user ratings on movies. Following the data preprocessing method of the previous work (Kang & McAuley, 2018; Sun et al., 2019; Li et al., 2023), we treat all reviews or ratings as implicit feedback (i.e., a user-item interaction), chronologically organize them by their timestamps and discard users and items with fewer than 5 related actions. The maximum sequence length is set to 200 for MovieLens-1M dataset, and 50 for the other two datasets. Besides, we adopt the *leave-one-out* evaluation strategy, leaving out the last item for test, the second-to-last item for validation, and the rest for training. The statistics of the processed datasets can be found in Table 1.

Table 1: **Statistics of three experimental datasets**

| Dataset | Beauty | Toys and Games | Movielens |
|---|---|---|---|
| **#Users** | 22,363 | 19,412 | 6,040 |
| **#Items** | 12,101 | 11,924 | 3,706 |
| **#Interactions** | 198,502 | 167,597 | 1,000,209 |
| **Avg. interactions per user** | 8.88 | 8.63 | 165.60 |
| **#Train Sequences** | 131,413 | 109,361 | 982,089 |

**Baselines** We evaluate SDREC against several representative sequential recommendation methods, including discriminative methods and generative methods. **GRU4Rec** (Hidasi et al., 2015), utilizes RNN to model the sequential behavior of users; **Caser** (Tang & Wang, 2018), devises horizontal and vertical CNN to exploit user's recent sub-sequence behaviors; **SASRec** (Kang & McAuley, 2018), utilizes a Transformer encoder to model the implicit correlations between items; **BERT4Rec** (Sun et al., 2019), proposes to adopt a bidirectional Transformer for recommendation; **STOSA** (Fan et al., 2022), adopts a stochastic Transformer with Wasserstein self-attention as sequence encoder; **SVAE** (Sachdeva et al., 2019), uses a variational self-attention network to characterize the uncertainty of user preferences; **ACVAE** (Xie et al., 2021), adopts an adversarial and contrastive variational autoencoder to learn personalized characteristics; **DreamRec** (Yang et al., 2024), generates the oracle item embeddings via diffusion model without discriminative learning; **DiffuRec** (Li et al., 2023), utilizes diffusion method to model users' multi-level interests; **DiffRec** (Wang et al., 2023), proposes to incorporate the diffusion model in collaborative filtering.

**Implementation Details** Follow the *full-ranking protocol* (He et al., 2020), we rank all the non-interacted items for each user. We evaluate all methods with two widely used metrics, H@K (Hit Rate) and N@K (Normalized Discounted Cumulative Gain), where $K = \{10, 20\}$. The code is implemented in Python 3.9 and PyTorch 1.10.0 and runs on NVIDIA P100 GPU with CUDA 11.8. We fix the learning rate as 0.001, batch size as 1024 and unconditional diffusion training probability $p_u$ as 0.4. We set the embedding dimension to 64 for the large dataset Movielens and 32 for the other two datasets. The number of diffusion steps is 32 and the noise schedule is linear across all datasets. More hyper-parameter settings and tuning range can be found in Appendix A.

## 4.2 RESULTS

**Overall Performance** Table 2 shows the overall results of our method against baseline models in terms of TopK recommendation. Compared to RNN-based method GRU4Rec and CNN-based method Caser, Transformer-based methods SASRec and BERT4Rec capture more complicated dependency relations and more complex item semantics, resulting in better recommendations. For generative approaches, diffusion-based methods DiffuRec, DiffRec and DreamRec achieve better performance

than VAE-based methods SVAE, ACVAE, and STOSA due to the strong fitting capabilities inherent in diffusion models. Meanwhile, our approach demonstrates notably enhanced performance across all datasets than all baselines. This superiority is attributed to our design, which effectively integrates item semantics into the diffusion process, allowing the denoising network to leverage the rich semantics embedded in item embeddings to accurately guide the generation direction. A visualized case provided in Figure 2 also demonstrates the strong ability of SDREC in leveraging the item semantics. We conducted additional evaluations on two more datasets. Due to space limitations, the results are provided in Appendix B.

Table 2: **Overall recommendation results on three datasets.** All results are reported in %. The best results are in **boldface**, and the second-best are in underlined. $^\star$ indicates the results are borrowed from (Li et al., 2023). "H" denotes *Hit Rate* while "N" denotes *Normalized Discounted Cumulative Gain*. SDREC was conducted three times with different random seeds.

| Algorithms | Beauty | | | | Toys and Games | | | | Movielens | | | |
|---|---|---|---|---|---|---|---|---|---|---|---|---|
| | H@10 | H@20 | N@10 | N@20 | H@10 | H@20 | N@10 | N@20 | H@10 | H@20 | N@10 | N@20 |
| GRU4Rec$^\star$ | 1.94 | 3.85 | 0.90 | 1.38 | 1.86 | 3.18 | 0.94 | 1.27 | 10.17 | 18.70 | 4.68 | 6.82 |
| Caser$^\star$ | 2.82 | 4.41 | 1.36 | 1.76 | 1.83 | 2.95 | 0.85 | 1.13 | 13.38 | 22.55 | 6.14 | 8.43 |
| SASRec$^\star$ | 6.27 | 8.98 | 3.23 | 3.66 | 6.55 | 9.23 | 3.75 | 4.33 | 16.89 | 28.32 | 7.73 | 10.60 |
| BERT4Rec$^\star$ | 3.72 | 5.79 | 1.83 | 2.35 | 2.93 | 4.59 | 1.49 | 1.90 | 20.57 | 29.95 | 11.13 | 13.48 |
| STOSA$^\star$ | 6.21 | 9.59 | 3.21 | 3.76 | 6.94 | 9.51 | 3.88 | 4.38 | 14.39 | 24.99 | 6.08 | 8.72 |
| SVAE$^\star$ | 1.98 | 3.15 | 0.99 | 1.29 | 1.36 | 1.92 | 0.71 | 0.85 | 2.72 | 5.03 | 1.23 | 1.83 |
| ACVAE$^\star$ | 3.88 | 6.12 | 2.14 | 2.70 | 3.08 | 4.41 | 1.85 | 2.18 | 19.93 | 28.97 | 10.54 | 12.82 |
| DreamRec | 4.32 | 6.14 | 2.84 | 3.03 | 4.74 | 5.32 | 3.23 | 3.38 | 20.66 | 27.60 | 12.28 | 14.03 |
| DiffRec | 6.25 | 8.51 | 3.55 | 4.14 | 6.57 | 8.68 | 3.88 | 4.41 | 11.84 | 19.93 | 6.06 | 8.11 |
| DiffuRec$^\star$ | 7.91 | 11.11 | 4.75 | 5.56 | 7.46 | 9.84 | 4.77 | 5.37 | 26.27 | 36.79 | 14.79 | 17.44 |
| SDREC | **8.62**±.33 | **11.81**±.32 | **5.27**±.19 | **6.07**±.21 | **9.45**±.18 | **12.34**±.18 | **6.12**±.15 | **6.85**±.23 | **32.38**±.77 | **42.83**±.43 | **18.89**±.55 | **21.51**±.28 |

**Ablation Study** To verify the effectiveness of each design choice of SDREC, we perform four ablation experiments, shown in Table 3. The removal of the contrastive loss (w/o discriminative learning) leads to a significant decline in results. This is mainly due to the lack of constraints on item representation learning results in inaccurate item semantics, and thus the direction of the denoising process becomes blurred. After replacing the `Semantic Fusion Layer` to a linear layer which simply reduces the dimension from $N$ to $D$ (w/o Semantic Fusion Layer), the performance drops substantially due to the lack of a comprehensive awareness of item semantics during the reverse denoising process. Hence, the `Semantic Fusion Layer` plays a crucial role in enhancing recommendation performance. Besides, the absence of unconditional training and sampling (w/o unconditional training and reweight sample) will induce overfitting, leading to a drop in recommendation performance. However, for Beauty dataset, whether unconditional training is introduced has little impact on performance. This is because we trained relatively fewer steps on this dataset (see Table 5), resulting in a less pronounced overfitting phenomenon.

Table 3: **Ablation results on three datasets.** All results are reported in %. The best results are in **boldface**. "H" denotes *Hit Rate* while "N" denotes *Normalized Discounted Cumulative Gain*.

| Settings | Beauty | | Toys and Games | | Movielens | |
|---|---|---|---|---|---|---|
| | H@10 | N@10 | H@10 | N@10 | H@10 | N@10 |
| **w/o discriminative learning** | 6.82 | 4.41 | 7.55 | 4.97 | 31.65 | 18.59 |
| **w/o Semantic Fusion Layer** | 6.27 | 3.48 | 7.46 | 4.53 | 20.65 | 11.16 |
| **w/o unconditional training** | 8.61 | **5.30** | 9.23 | 5.96 | 31.84 | 18.60 |
| **w/o reweight sample** | 8.43 | 5.25 | 9.33 | 6.03 | 31.92 | 18.49 |
| original | **8.62** | 5.27 | **9.45** | **6.12** | **32.38** | **18.89** |

**Impact of Hyper-parameters** We evaluate the recommendation results of SDREC on different diffusion steps, noise schedules, and unconditional training probabilities $p_u$. Figure 4 shows the results for Beauty dataset. As we can see, small diffusion steps (i.e., 8) notably hurt the performance. As the diffusion steps increase (i.e. $\geq 16$), we observe a discernible improvement in performance, while more diffusion steps won't have much impact. As for the noise schedules, we observe that the

linear schedule consistently yields the most favorable results, while the truncated linear and square root schedules offer slightly worse performance compared to the linear schedule. Conversely, the cosine and truncated cosine schedules have exhibited notably inferior results in our experiments. Finally, our experiments show that setting unconditional training probabilities $p_u$ to 0.4 achieves the best recommendation performance. The other two dataset results are in Appendix C.

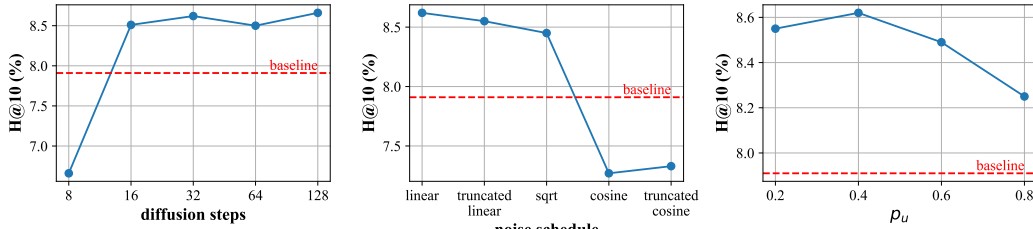

Figure 4: **Impact of some hyper-parameters.** Results for SDREC on **Beauty** dataset with different diffusion steps, noise schedules, and unconditional training probabilities $p_u$. "H" denotes *Hit Rate*. Baseline is **DiffuRec** (Li et al., 2023).

**Inference Efficiency**  Due to the latency limitation for online systems, the inference speed is crucial for recommenders. Therefore, we compare the total inference time of three state-of-the-art diffusion recommenders on the test split of three datasets, shown in Table 4. For a fair comparison, we set the batch size to 1024 for all datasets and algorithms. Note that the batch size for DiffuRec on Movielens is set to 512 due to the GPU memory limitations. DreamRec takes the longest inference time due to its large embedding dimensionality (i.e., $> 1024$) and extensive diffusion steps (i.e., $> 500$) required for favourable results. In contrast, DiffuRec employs a much smaller embedding dimensionality (i.e., 128) and achieves superior inference speed. DiffRec applies a smaller network (i.e., MLP) compared to the four-layer Transformer encoder of DiffuRec, further reducing inference time. Our method accelerates the inference even more by utilizing `Semantic Fusion Layer`, reducing the $N$-dimensional distribution vector to a $D$-dimensional vector before passing it through the denoising network. This reduction significantly cuts down computational complexity.

Table 4: **Inference time on test split of three datasets.** The best results are in **boldface**.

| Algorithms | Beauty | Toys and Games | Movielens |
|:---:|:---:|:---:|:---:|
| **DreamRec** (Yang et al., 2024) | 283.42s | 239.70s | 74.37s |
| **DiffRec** (Wang et al., 2023) | 10.33s | 9.35s | 1.47s |
| **DiffuRec** (Li et al., 2023) | 82.47s | 70.34s | 113.31s |
| **SDREC** | **7.51s** | **6.37s** | **1.31s** |

## 5  CONCLUSION AND LIMITATIONS

In this paper, we propose SDREC, a semantic-aware diffusion model for sequential recommendation which can efficiently leverage the item semantics during the diffusion process. Inspired by the attention mechanism, we designed the `Semantic Fusion Layer`. In this layer, the embedding table is weighted by the noisy input distribution, allowing the reverse denoising process aware of the item semantics comprehensively. Combined with contrastive learning, which constraints the embedding table to learn discriminant information, SDREC will better extract item semantics contained in the embeddings. Experiments demonstrate promise gain compared with existing methods. Furthermore, as SDREC shows efficient inference speed, it is friendly to online services.

However, there is still a limitation for SDREC. The current model structure is based on a fixed candidate set, which is not suitable for handling new items in real recommendation scenarios. We believe that advanced methods for cold start scenarios will mitigate this problem, which also provides new research opportunities for future work.

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

## A    SETTINGS OF HYPER-PARAMETERS FOR SDREC

We tuned the learning rate across the range of [0.0001, 0.001, 0.005, 0.01]; the embedding dimension was explored within [16, 32, 64, 128]; the number of encoding layers were tuned between [1, 2, 4]; the number of diffusion steps were explored in the range of [8, 16, 32, 64, 128]; the diffusion noise schedule options included [linear, cosine, sqrt, truncated linear, truncated cosine]; and the unconditional training probability was tested within [0, 0.2, 0.4, 0.6, 0.8].

The final hyper-parameter settings of SDREC for the three experimental datasets are shown in Table 5.

Table 5: **Hyper-parameter settings of SDREC for three experimental datasets**

| Dataset | Beauty | Toys and Games | Movielens |
|---|---|---|---|
| **learning rate** | 0.001 | 0.001 | 0.001 |
| **training steps** | 77400 (600 epochs) | 107000 (1000 epochs) | 480000 (500 epochs) |
| **batch size** | 1024 | 1024 | 1024 |
| $p_u$ | 0.4 | 0.4 | 0.4 |
| **#encoder layers** | 2 | 1 | 2 |
| **#attention heads** | 4 | 4 | 4 |
| **hidden dimension** | 32 | 32 | 64 |
| **dropout ratio** | 0.3 | 0.5 | 0.3 |
| **diffusion steps** | 32 | 32 | 32 |
| **noise schedule** | linear | linear | linear |

## B    RECOMMENDATION PERFORMANCE ON MORE DATASETS

Table 6: **Recommendation results on two more datasets.** The best results are in **boldface**, and the second-best are underlined. $\star$ indicates the results are borrowed from (Yang et al., 2024). "H" denotes *Hit Rate* while "N" denotes *Normalized Discounted Cumulative Gain*.

| Algorithms | YooChoose | | Zhihu | |
|---|---|---|---|---|
| | H@20(%) | N@20(%) | H@20(%) | N@20(%) |
| **GRU4Rec**$\star$ (Hidasi et al., 2015) | 3.89±.11 | 1.62±.02 | 1.78±.12 | 0.67±.03 |
| **Caser**$\star$ (Tang & Wang, 2018) | 4.06±.12 | 1.88±.09 | 1.57±.05 | 0.59±.01 |
| **SASRec**$\star$ (Kang & McAuley, 2018) | 3.68±.08 | 1.63±.02 | 1.62±.01 | 0.60±.03 |
| **DreamRec**$\star$ (Yang et al., 2024) | 4.78±.06 | 2.23±.02 | 2.26±.07 | 0.79±.01 |
| **DiffRec**$\star$ (Wang et al., 2023) | 4.33±.02 | 1.84±.01 | 1.82±.03 | 0.65±.09 |
| **DiffuRec** (Li et al., 2023) | 4.72±.10 | 2.40±.05 | 1.58±.15 | 0.58±.05 |
| **SDREC** | **4.92±.08** | **2.54±.02** | **2.52±.14** | **0.91±.04** |

To fully test the performance of SDREC, we additionally evaluate the recommendation results on two more datasets: 1) **YooChoose** from RecSys Challenge 2015 (on Recommender Systems, 2015) and we use the purchase sequences of the medium size data; 2) **Zhihu** (Hao et al., 2021) which is collected from a socialized knowledge Q&A platform. These two datasets are processed and split according to (Yang et al., 2024).

We train SDREC on YooChoose dataset for 300 epochs, with $p_u = 0.2$ and truncated linear schedule. Other hyper-parameters remain consistent with those of Beauty dataset as Table 5 shows. For Zhihu dataset, we adopt the same hyper-parameters used for Toys and Games dataset except that the learning rate is set to 0.0005, training epochs are set to 300 and the batch size is set to 256.

The recommendation results for the above two datasets are reported in Table 6. As demonstrated, SDREC outperforms all baselines, showing the high efficiency of our design.

## C  MORE RESULTS FOR THE IMPACT OF HYPER-PARAMETERS

Figure 5 and Figure 6 illustrate the impact of some hyper-parameters for SDREC on Toys and Games and Movielens datasets.

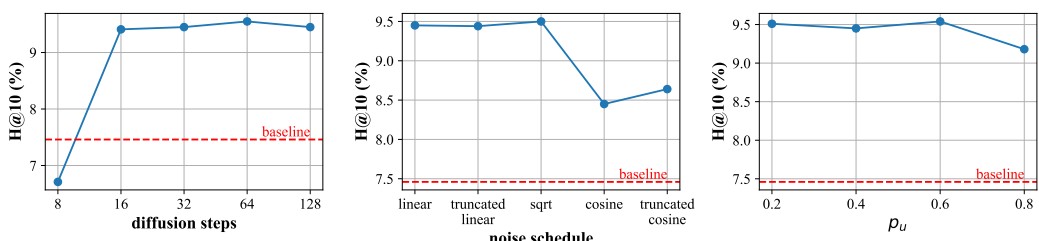

Figure 5: **Impact of some hyper-parameters.** Results for SDREC on **Toys and Games** dataset with different diffusion steps, noise schedules, and unconditional training probabilities $p_u$. "H" denotes *Hit Rate*. Baseline is **DiffuRec** (Li et al., 2023).

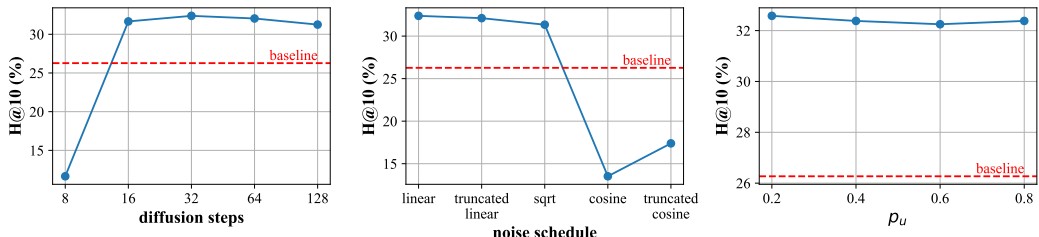

Figure 6: **Impact of some hyper-parameters.** Results for SDREC on **Movielens** dataset with different diffusion steps, noise schedules, and unconditional training probabilities $p_u$. "H" denotes *Hit Rate*. Baseline is **DiffuRec** (Li et al., 2023).

The choice of unconditional training probability $p_u$ does not exert a significant impact on these two datasets. This is mainly because of the large number of training steps for these two datasets as Table 5 shows. Consequently, even if $p_u$ is increased, the sufficient number of conditional training steps ensures the attainment of favorable results.

