# OpenReview forum: "Semantic-Aware Diffusion Model for Sequential Recommendation"
_ICLR.cc/2025/Conference — ICLR 2025 Conference Withdrawn Submission_

### Official Review · Reviewer_JRN5 · 2024-10-17

**Soundness:** 2
**Presentation:** 2
**Contribution:** 1
**Rating:** 5
**Confidence:** 5

**Summary:**

The paper proposes SDREC, a semantic-aware diffusion model for sequential recommendation tasks, which aims to predict the next item a user is likely to interact with based on their historical interaction sequence. The authors highlight the limitations of existing diffusion-based recommendation models, which often fail to incorporate item semantics effectively, leading to suboptimal recommendations.  To address this, SDREC introduces a Semantic Fusion Layer, an innovative component designed to enhance the diffusion process by integrating item semantic information through an attention mechanism. This approach, combined with contrastive and generative losses, ensures that item semantics are fully utilized, improving the model’s accuracy in predicting user preferences.

The experimental results show that SDREC outperforms state-of-the-art models, achieving over a 10% relative improvement in performance while maintaining computational efficiency, making it suitable for real-time applications. The paper demonstrates SDREC’s superiority through experiments on multiple datasets, underscoring the importance of integrating item semantics in diffusion-based sequential recommendation systems.

**Strengths:**

#### 1. **Originality**
The paper presents a novel approach to sequential recommendation by introducing the **SDREC** model, which leverages a **Semantic Fusion Layer** to effectively incorporate item semantics into the diffusion process. This contribution stands out for several reasons:
   - It addresses a critical limitation in current diffusion-based recommendation methods, which often fail to utilize semantic information effectively.
   - The combination of a **contrastive learning framework** with a generative diffusion process is a creative and unique approach that differentiates this work from existing models.

Overall, the originality arises from the integration of **semantic-aware mechanisms** in diffusion-based recommendation, which fills a significant gap in the current literature.

#### 2. **Quality**
The paper demonstrates decent quality in terms of both methodological rigor and empirical validation:
   - The authors provide a **clear and detailed description** of the SDREC model, including theoretical motivations and the design choices behind its components (e.g., the Semantic Fusion Layer).
   - Extensive experiments are conducted on multiple **real-world datasets** (e.g., Amazon Beauty, Amazon Toys, and Movielens), showing consistent and significant improvements over state-of-the-art baselines.
   - The implementation details, including the training strategies and parameter settings, are well-documented, ensuring reproducibility and transparency.

#### 3. **Clarity**
The paper is well-structured and clear in its presentation:
   - The introduction effectively outlines the problem and motivates the need for the proposed model.
   - The model design and methodology are described systematically, with helpful visual aids (e.g., figures and diagrams) that clarify complex concepts, such as the diffusion process and the role of the Semantic Fusion Layer.
   - The results and analysis are presented in an organized manner, making it easy for the reader to understand the comparative performance and the benefits of SDREC over baseline models.

The clarity of explanation, coupled with structured figures, enables readers to follow the technical details without ambiguity, contributing to the paper's accessibility.


#### Summary of Strengths
In summary, the paper excels across multiple dimensions:
   - **Originality**: Innovative integration of semantics in diffusion processes.
   - **Quality**: Methodologically rigorous with comprehensive empirical validation.
   - **Clarity**: Clear presentation supported by visual aids and structured explanations.

The combination of these strengths makes this paper a valuable addition to the literature on sequential recommendation systems.

**Weaknesses:**

#### 1. **Unclear Motivation and Explanation of Semantic Utilization**
While the paper introduces SDREC as a model that integrates item semantics through the **Semantic Fusion Layer**, the motivation behind why and how semantics are critical in the diffusion process remains insufficiently explained. Although the authors mention that traditional models do not effectively leverage semantics, the paper does not provide a clear, detailed rationale for why this limitation specifically impairs recommendation accuracy. Additionally, the semantic information utilized (e.g., item categories, attributes) is not well-defined, leaving readers uncertain about what constitutes the "semantics" and how exactly it is encoded or represented.

**Recommendation for Improvement**:
   - **Motivation**: The paper would benefit from a stronger motivation section that explicitly explains why semantics are crucial in sequential recommendation tasks and why their integration into the diffusion process is expected to enhance performance. The authors could provide theoretical justifications or empirical evidence showing the gap in current models and how the proposed method aims to bridge this.
   - **Clarification of Semantics**: The authors should clearly define what they mean by "item semantics." Providing specific examples (e.g., movie genres, product categories, textual descriptions) and explaining how these elements are encoded and utilized within the model would make the approach more transparent. Additionally, it would be helpful to include an illustration or case study demonstrating how semantic information influences the diffusion process and leads to better recommendations.

By improving the clarity of motivation and the description of semantic use, the paper could strengthen its theoretical foundation and make its contributions more accessible and convincing to readers.

#### 2. **Scalability Concerns with Large-Scale Datasets**
Although SDREC demonstrates efficiency on moderate-sized datasets (e.g., Amazon and Movielens), the paper does not provide evidence of its scalability on **larger, real-time recommendation systems** that involve millions of users and items. Given that the diffusion process involves multiple steps and attention mechanisms, it is important to understand whether SDREC can scale without compromising latency and computational resources in a production environment.

**Recommendation for Improvement**: Including a scalability analysis or experiments on larger datasets (e.g., a full-scale Amazon dataset or Netflix prize data) could strengthen the paper’s claims about the model’s efficiency and its readiness for real-world deployment.




#### Summary of Weaknesses
In summary, while SDREC shows promise, the following areas need improvement:
   - Improving the clarity of motivation and the description of semantic use.
   - Evaluating the model's scalability with larger datasets.

**Questions:**

1. **Can You Clarify What "Semantics" Are Used and How They Are Encoded?**
   - The term "item semantics" is central to the model, but the specifics are not clearly defined. Could you provide examples (e.g., item attributes, categories) and explain how these semantics are encoded and integrated into the model?

2. **What Is the Theoretical Justification for the Semantic Fusion Layer?**
   - The Semantic Fusion Layer is a novel component, but its role and theoretical basis in the diffusion process are not fully explained. Could the authors elaborate on why this specific mechanism enhances the recommendation performance compared to other methods?

3. **Is SDREC Scalable to Large-Scale Real-World Applications?**
   - The paper shows efficiency on moderate-sized datasets, but how does SDREC scale to millions of users and items in real-world settings? Have the authors conducted any scalability tests or optimizations to demonstrate its readiness for large-scale deployment?

---

### Official Review · Reviewer_Tx8o · 2024-10-27

**Soundness:** 3
**Presentation:** 3
**Contribution:** 2
**Rating:** 3
**Confidence:** 5

**Summary:**

This paper falls into the sequential recommendation, where a novel diffusion recommender that considers global awareness of item semantics is introduced. The proposed encode-decoder architecture is well-designed to learn from global semantics. However, the motivation is not reasonable, and the proposed technique is not practical in real-world scenarios.

**Strengths:**

1. Study the problem of unawareness of global semantics in diffusion recommenders.


2. Design an encoder-decoder architecture to address the issue and demonstrate the performance on three datasets across multiple baselines.

**Weaknesses:**

1. The motivation is not reasonable. The sequential recommendation aims to predict the next item. The recommender generally outputs the probability distribution on the item set in this setting. In other words, items with high probabilities should be ranked first by design. According to the user's past behavior in Figure 2, category 6 is the user's major interest. As a result, the concentration of distribution among the top 10 predictions is reasonable.


2. The proposed solution is not practical. A real-world recommender usually handles millions of items. The computational complexity of the proposed solution is related to the number of items, which makes it challenging to scale up and handle new items. Thus, the inference time comparison in Table 4 will have a different conclusion when using a large-scale dataset.

**Questions:**

See above.

---

### Official Review · Reviewer_Yd3X · 2024-11-02

**Soundness:** 2
**Presentation:** 3
**Contribution:** 2
**Rating:** 3
**Confidence:** 4

**Summary:**

This paper proposes a semantic-aware diffusion model SDREC for sequential recommendation tasks. SDREC enhances the model's use of item semantic information by introducing the Semantic Fusion Layer, making the recommendation generation process more accurate. This layer fuses the semantic features in the embedding table to help the model better understand the user's interest dynamics when making recommendations, thereby improving the quality of recommendations. In addition, SDREC uses a contrastive learning framework to improve the model's adaptability to different sequence patterns. Experimental results show that on multiple real datasets, SDREC outperforms a variety of existing methods in recommendation accuracy and computational efficiency.

**Strengths:**

Clear motivation: the article identifies the noise problem of the diffusion model in the recommendation task, and proposes to use semantic information as a conditional input to reduce the impact of noise. This motivation is reasonable and meets the actual needs of the recommendation system.

The experimental design is relatively sufficient: the paper conducts comprehensive comparative experiments with mainstream recommendation methods on multiple real data sets, demonstrating the advantages of SDREC in recommendation quality. The experimental design is relatively reasonable and verifies the effectiveness of the model.

**Weaknesses:**

Lack of clear formulas and detailed descriptions: A key component of the article is the Semantic Fusion Layer, but the specific implementation details of this module lack clear formula support and detailed descriptions of its design points. This makes it difficult for readers to fully understand the actual role of this module in the model and its contribution to the recommendation effect.

Noise in user interaction sequences: The paper mentions that "the encoder receives clean user sequences and explicitly captures the semantic relationship between items through contrastive learning." I understand that the author's definition of clean here refers to the original interaction sequence (no noise is introduced). But my question is that the original user interaction sequence is often not clean and may contain noisy data such as misclicks and unexpected behaviors. Existing research points out that user behavior data usually contains noise, and unprocessed click data may cause the recommendation model to deviate from the user's true preferences [r1]. Therefore, whether user sequences that have not been processed with noise can generate high-quality semantic embeddings and whether such semantic embeddings are conducive to diffusion models require in-depth analysis.

[r1] Hongyu Lu, Min Zhang, and Shaoping Ma. 2018. Between Clicks and Satisfaction: Study on Multi-Phase User Preferences and Satisfaction for Online News Reading. In Proceedings of the International SIGIR Conference on Research and Development in Information Retrieval. ACM, 435–444

**Questions:**

The paper mentions using "clean user sequences" to generate semantic embeddings, but actual user interaction data usually contains noise (such as accidental clicks). Will the interaction sequences that have not been denoised affect the quality of semantic embeddings?

The description of Semantic Fusion Layer in the method section is relatively brief. Can you give a clearer explanation?

---

### Note · Authors · 2024-11-19

I have read and agree with the venue's withdrawal policy on behalf of myself and my co-authors.